

# Seq2science: an end-to-end workflow for functional genomics analysis

Maarten van der Sande*, Siebren Frölich*, Tilman Schäfers, Jos G.A. Smits, Rebecca R. Snabel, Sybren Rinzema and Simon J. van Heeringen

Molecular Developmental Biology, Radboud University Nijmegen, Nijmegen, the Netherlands
* These authors contributed equally to this work.

## ABSTRACT

Sequencing databases contain enormous amounts of functional genomics data, making them an extensive resource for genome-scale analysis. Reanalyzing publicly available data, and integrating it with new, project-specific data sets, can be invaluable. With current technologies, genomic experiments have become feasible for virtually any species of interest. However, using and integrating this data comes with its challenges, such as standardized and reproducible analysis. Seq2science is a multi-purpose workflow that covers preprocessing, quality control, visualization, and analysis of functional genomics sequencing data. It facilitates the downloading of sequencing data from all major databases, including NCBI SRA, EBI ENA, DDBJ, GSA, and ENCODE. Furthermore, it automates the retrieval of any genome assembly available from Ensembl, NCBI, and UCSC. It has been tested on a variety of species, and includes diverse workflows such as ATAC-, RNA-, and ChIP-seq. It consists of both generic as well as advanced steps, such as differential gene expression or peak accessibility analysis and differential motif analysis. Seq2science is built on the Snakemake workflow language and thus can be run on a range of computing infrastructures. It is available at https://github.com/vanheeringen-lab/seq2science.

## INTRODUCTION

The Sequence Read Archive (SRA) at NCBI currently holds over 36 petabytes of sequencing data, and this volume is growing rapidly (*NCBI insights, 2020*). Due to the flexibility of using sequencing as a readout, a large variety of different assays are available, such as RNA-sequencing (RNA-seq) (*Nagalakshmi et al., 2008*) for gene expression quantification, Chromatin Immunoprecipitation (ChIP) sequencing (*Johnson et al., 2007*) to profile DNA-bound proteins and assay for transposase-accessible chromatin with sequencing (ATAC-seq) (*Buenrostro et al., 2015*) to determine DNA accessibility. This wealth of public data enables researchers to verify results, re-analyze data with novel techniques, and to combine and integrate datasets from different studies. However, processing these large amounts of data is a challenging and time-consuming task, even for researchers that are already familiar with high-throughput sequencing data processing

Corresponding authors
Maarten van der Sande, sande@science.ru.nl
Simon J. van Heeringen, simon.vanheeringen@gmail.com

details. To address this issue, various workflow systems have been developed, roughly categorized into three approaches: community-oriented workflow collections, multi-purpose workflows, and single-purpose workflows.

Community-oriented workflow collections enable multiple users to contribute workflows, as long as they conform to the established style and language of the community. Examples of community-based workflow collections include Galaxy (*The Galaxy Community, 2022*), Snakemake-Workflows (*Snakemake workflows, 2023*), and nf-core (*Ewels et al., 2020*). These collections offer the advantage of supporting a wide range of workflows and assays, with an active community providing support. Multi-purpose workflows facilitate multiple highly consistent workflows and typically provide a single entry point for users. Examples include Snakepipes (*Bhardwaj et al., 2019*), ENCODE pipelines (*Hitz et al., 2023*), and CellRanger (*Zheng et al., 2017*). These workflows are designed to maintain consistency across different analyses, making it easier for users to learn and analyze the supported workflows. Single-purpose workflows are tailored to address specific problems, such as ARMOR for RNA-seq (*Orjuela et al., 2019*) and PEPATAC to analyze ATAC-seq (*Smith et al., 2021*). The advantage of these workflows lies in their high level of specialization, focusing on particular tasks or analyses.

Although there is a choice of publicly available, published workflows, we found that these did not address all of our requirements. First, apart from some exceptions such as Galaxy, most workflows have not been specifically designed with public data in mind. This requires users to download and prepare the data in advance. While this is doable for small studies, it quickly becomes prohibitive for more large-scale analyses combining data from different studies. Additionally, many workflows have been primarily developed for, and tested on, human and/or mouse data. This limits their applicability to non-model species. It can be cumbersome to add new genomes and supporting gene and transcript annotation. Finally, workflows that do not actively encourage data exploration leave users susceptible to missing biases and failing to uncover concealed insights. Workflows should make sure to include a variety of quality control results and diagnostic plots, as it is essential to check the quality of the data. This includes, for instance, checking mapped data visually using a genome browser.

To address these limitations, we have developed seq2science, a multi-purpose workflow that supports virtually all public sequencing databases, multiple sequencing assays, and any species of interest. Seq2science is capable of automatically downloading genome assemblies and raw sequencing data from a range of sources. It supports multiple read trimmers, aligners, peak callers, and quantification methods, and generates an extensive quality report and a fully configured UCSC trackhub. It currently supports bulk ChIP-, ATAC- and RNA-seq, downloading of FASTQ files, and a generic genomic alignment workflow. Installation is easy through the Conda package manager, and extensive documentation is available online (https://vanheeringen-lab.github.io/seq2science/). Seq2science is designed to cater to both intermediate and advanced bioinformaticians. It serves as an accessible starting point for those with a basic grasp of bioinformatics

concepts, thanks to its sensible default settings. Additionally, it offers a high degree of customization, making it appealing to advanced users who seek more tailored control over their analyses.

# METHODS

## Implementation

Seq2science is built using Snakemake (*Mölder et al., 2021*), a portable and open-source workflow system, which divides a workflow into independent modules called rules. Each rule includes a piece of code, its expected output, and optional input requirements. This design allows rules to be linked together, with the output of one rule serving as the input for another. Snakemake automatically determines the order in which rules need to be executed and distributes these tasks across available resources. To ensure reproducibility, most rules are assigned a specific virtual environment using the Conda ecosystem (https://www.anaconda.com/download) that is automatically installed at the start of a run.

Seq2science requires two input files: a samples file and a configuration file. The samples file is a table containing a column of FASTQ files (or public identifiers for automatic downloading from any of the supported databases), a column with the assembly the FASTQ file must be mapped to, and optional additional metadata columns. Optional columns are the descriptive name of the samples, information about the relations between samples such as whether samples are technical and/or biological replicates, and other such details. The configuration file is a YAML file with configurable parameters. These include whether to execute certain steps, the options to use when executing rules, and the directories where the output will be stored. Detailed explanations for the samples file and configuration file are available in the online documentation, and examples of these files are in the Supplemental Information and available at Zenodo (https://doi.org/10.5281/zenodo/8345208).

### General overview

Regardless of the chosen workflow, when seq2science is executed, it checks for the local availability of the genome assembly and sequencing reads (in FASTQ format). If the genome assembly or the reads are not found locally, seq2science will download them. Once the raw data is obtained, the FASTQ files are prepared for alignment. This involves building a genome index and trimming the reads. The aligned reads are then duplicate-marked, filtered, and sorted according to the configuration settings. Nearly all workflows produce a set of indexed BAM (or optionally CRAM) files, an extensive quality control report, and a UCSC trackhub at this stage. The ATAC- and ChIP-seq workflows call peaks, and the output is stored and aggregated into a peak counts table. The RNA-seq workflow uses the specified quantifier to obtain raw gene counts and TPM tables. Optional differential gene expression, peak accessibility, and motif activity analyses are fully supported. Finally, a workflow explanation is generated with the parameters, version, and citation per tool which is also embedded in the QC report. For a schematic overview of the different steps and supported tools of seq2science see Fig. 1.

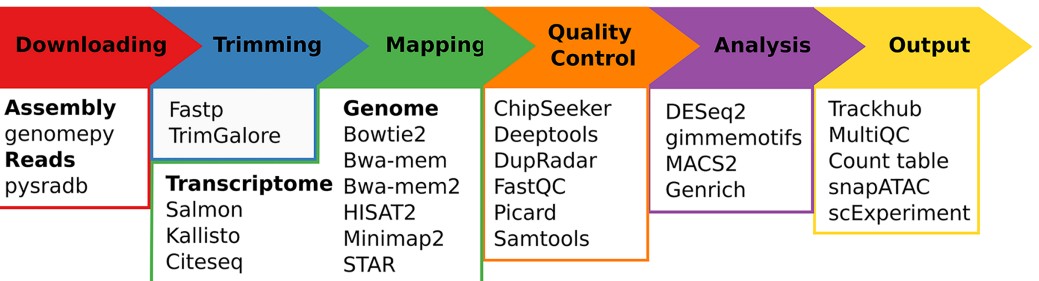

**Figure 1 Schematic overview of seq2science.** Seq2science can conceptually be split into six parts: downloading of samples and genome assembly, trimming of reads, transcriptome/genome mapping, quality control, initial analysis, and the final output. For each part the corresponding supported tools are listed.

### Download-fastq

The download-fastq workflow can retrieve FASTQ files from various databases: the European Nucleotide Archive (ENA) (*Leinonen et al., 2010*), Gene Expression Omnibus (GEO) (*Barrett et al., 2012*), the Sequence Read Archive (SRA) (*Leinonen, Sugawara & Shumway, 2010*), the DNA Data Bank of Japan (DDBJ) (*Kodama, Shumway & and, 2011*), the Genome Sequence Archive (GSA) (*Wang et al., 2017*) and the ENCODE project (*Luo et al., 2019*), using their specific identifiers; ERX, ERR, GSM, SRX, SRR, DRX, DRR, CRX, ENCSR, or ENCFF numbers. The SRA, ENA, and DDBJ databases contain raw sequencing data that must be converted to FASTQ format, and they generally mirror each other in their content. EBI ENA, GSA, and ENCODE, however, store FASTQ files directly, so if a sample on the SRA or DDBJ is found to be mirrored on ENA by pysradb (*Choudhary, 2019*), seq2science will directly download FASTQ files from there, optionally using Aspera Connect (ascp), which is a high-speed transfer protocol developed by IBM. If the sample is not directly available in FASTQ format, seq2science uses the sra-toolkit (*Leinonen, Sugawara & Shumway, 2010*) to download the raw data and parallel-fastq-dump, a parallelized version of fastq-dump, to convert the data to FASTQ files.

### Alignment

The alignment workflow in seq2science processes FASTQ files that are either already present on the device or are automatically obtained using the download-fastq workflow. If necessary, the workflow will also download a genome assembly FASTA file and corresponding gene annotation file using genomepy (*Frölich et al., 2023*). The FASTQ files are trimmed for quality and adapters using either TrimGalore (*Krueger et al., 2023*) or fastp (*Chen et al., 2018*), as specified by the user. The trimmed FASTQ files are then aligned to the genome using the selected mapper, such as bowtie2 (*Langmead & Salzberg, 2012*), bwa-mem (*Li, 2013*), bwa-mem2 (*Vasimuddin et al., 2019*), HISAT2 (*Kim et al., 2019*), minimap2 (*Li, 2021*), or STAR (*Dobin et al., 2012*). The resulting BAM file is filtered based on criteria such as MAPQ value, duplicate status, or alignment in the ENCODE blacklist (*Amemiya, Kundaje & Boyle, 2019*). The filtered BAM file is then converted into a bigWig file and prepared for visualization in a UCSC trackhub (*Kent et al., 2002*) or, when the

genome assembly is not hosted by UCSC, as an assembly hub. The filtered BAM file can optionally be stored as a CRAM file to save disk space. Quality checks are performed throughout the process using FastQC (*Andrews, 2010*), samtools (*Li et al., 2009*), Picard (*Broad Institute, 2023*), and deepTools (*Ramírez et al., 2014*), and the results are summarized in a MultiQC report (*Ewels et al., 2016*).

### ATAC- & ChIP-seq

The ATAC- and ChIP-seq workflows are identical in implementation, except that they are initialized with different default settings. They internally use the same rules as the alignment and download-fastq workflows, which means that they either start by downloading FASTQ files, or analyze files that are already present. For the ATAC-seq workflow the aligned reads are by default Tn5 shifted (*Yan et al., 2020*) and only reads with a maximum template length of 150 base pairs are kept. Peak calling is done on the filtered BAM files with either MACS2 (*Zhang et al., 2008*) or genrich (*Gaspar, 2023*), with optionally specified control samples. Biological replicates can be combined either with the internal Fisher's method of either tool, or with IDR (*Li et al., 2011*). Peaks between conditions are combined when they fall within a certain range of each other with GimmeMotifs (*Bruse & van Heeringen, 2018*) and a count table is made for all samples based on the number of reads in peaks. Optional differential peak analysis with DESeq2 (*Love, Huber & Anders, 2014*), or differential motif analysis with GimmeMotifs (*Bruse & van Heeringen, 2018*) can be performed if selected. When doing a differential motif analysis, seq2science automatically converts the transcription factor gene names in the motif database into the (orthologous) gene names of the assembly used when a genome annotation is available. Additional QC is collected by deepTools (*Ramírez et al., 2014*), ChIPseeker (*Yu, Wang & He, 2015*), and Subread (*Liao, Smyth & Shi, 2013*). See File S14 for a directed acyclic graph of all the steps involved with the ATAC- and ChIP-seq workflows.

### RNA-seq

The RNA-seq workflow begins with the acquisition and processing of FASTQ files as described in the alignment and download-fastq workflows. Gene expression quantification can be based on either genomic alignment or transcript quantification, depending on the settings. For genomic alignment, reads are aligned to the genome with a splice-aware aligner (STAR (*Dobin et al., 2012*) or HISAT2 (*Kim et al., 2019*)). The output BAM files are filtered and have their duplicate reads marked. Gene expression quantification is then performed by assigning reads to genes, using HTSeq (*Anders, Pyl & Huber, 2014*) or featureCounts (*Liao, Smyth & Shi, 2013*). Ambiguous transcripts are minimized by providing the gene counting tools with the strandedness of each sample, which is inferred using RSeQC (*Wang, Wang & Li, 2012*). Additional gene-based TPM expression levels are generated using genomepy (*Frölich et al., 2023*), based on longest transcript lengths. For the gene quantification approach, transcript abundances are quantified using Salmon (*Patro et al., 2017*) in mapping-based mode. To improve mapping accuracy, decoy

sequences are generated as suggested by the Salmon documentation. The transcript abundances are aggregated to gene level using pytxi (*Frölich & van Heeringen, 2023*) or tximeta (*Love et al., 2019*) and additionally converted to gene counts using genomepy (*Frölich et al., 2023*).

Independent of the configured gene expression quantification approach, the workflow supports differential gene expression analysis with DESeq2 (*Love, Huber & Anders, 2014*) with batch effect correction to integrate (multiple) datasets, and can prepare an exon count table for downstream use with DEXSeq (*Anders, Reyes & Huber, 2012*). Strand-specific bigWig files are generated for visualization in a UCSC trackhub. The trackhub configuration file is updated with strand information for ease of use. Additional quality control metrics specific to RNA-seq are obtained from DESeq2 (*Love, Huber & Anders, 2014*) and dupRadar (*Sayols, Scherzinger & Klein, 2016*). See File S15 for a directed acyclic graph of all the steps involved with the RNA-seq workflow.

## USE CASES

To briefly illustrate some of the capabilities of seq2science, we show how to use it to download publicly available FASTQ files, and finally show three example processing runs using public data.

### Downloading FASTQ files

Downloading FASTQ files with seq2science has been made extremely easy. After installation of seq2science (see File S1), all that needs to be provided is a tab-separated file with database identifiers. Seq2science currently supports identifiers for ENA, GEO, SRA, DDBJ, GSA, and ENCODE. For this example, we will download one FASTQ file from each database. To get started we need to initialize seq2science in the current directory:

```
seq2science init download-fastq
```

After this initialization, we get a config and a samples file. The config file is practically empty as there is not much to configure for the download-fastq workflow. For this workflow, the relevant option is the directory where we want the samples to be downloaded.

We then edit the samples file and add the experiment (or run identifier) of each sample (Table 1). Here we show a mixture of paired-end and single-end samples to highlight seq2science's ability to work with both data types. To download these samples all we now have to do is run seq2science:

```
seq2science run download-fastq --cores 8
```

Seq2science will now start downloading our samples. Samples hosted on GSA, ENCODE, and ENA are downloaded directly as FASTQ file, whilst the samples not on ENA, ENCODE, or GSA first get downloaded as an intermediate SRA file which then gets converted into a FASTQ file. Even though we have specified the DRX and DRR samples by their DDBJ identifier, seq2science finds their ENA mirror if it exists and directly downloads those to save computational resources. At the end of this run, we end up with the corresponding FASTQ files for all 10 samples.

**Table 1 Example samples file for the download-fastq workflow.**

| Sample |
| --- |
| ERX000401 |
| ERR022487 |
| GSM2811115 |
| SRX257149 |
| SRR800037 |
| DRX029591 |
| DRR032791 |
| CRX269079 |
| ENCSR535GFO |
| ENCFF172MDS |

## A map of cis-regulatory elements in zebrafish

In this example, we reproduced part of the analysis of the study "A map of cis-regulatory elements and 3D genome structures in zebrafish" (*Yang et al., 2020*). *Yang et al. (2020)* studied zebrafish chromosome conservation with a wide array of different functional genomics techniques. Here, we focused on the analysis of cis-regulatory elements and gene expression in different embryonic tissues. Using the default tools incorporated into seq2science, we downloaded the raw data for the different assays from the SRA, aligned these data to the zebrafish genome (Fig. 2A) and performed a differential transcription factor motif enrichment analysis on the ATAC-seq data.

After running seq2science, we obtained a set of aligned BAM files, narrowPeak files, a count table, a trackhub, and a quality control report (see Files S2–S7 for the configuration, samples, and QC report). Figure 2A shows the alignment of reads visualized using the UCSC trackhub that was created by seq2science. The figure shows histone modification ChIP-seq data (H3K4me3, H3K27ac, H3K9me2 and H3K9me3), ATAC-seq, and RNA-seq for three different tissues (brain, muscle and liver) on a region of chromosome 4. Figures 2B and 2C show a selection of diagnostic plots from the quality report. Figure 2B demonstrates the ratio of ATAC-seq peaks annotated to different genomic regions, created by ChIPseeker. Figure 2C shows the correlation of reads in ATAC-seq peaks between samples. This figure illustrates the value of extensive quality control, as it shows two samples where the replicates do not cluster together (colon and intestine). This would warrant further investigation, as it could indicate a potential sample swap. In our experience, this occurs frequently with samples downloaded from public databases, which may be due to a sample swap in the original analysis, or during submission to the repository.

As a demonstration of a more high-level analysis, Fig. 3 shows the result of the differential motif analysis. Here, seq2science used GimmeMotifs to automatically convert the transcription factors in the motif database into the orthologous zebrafish genes. This automatic assignment means that motif analysis can also be used for non-model species

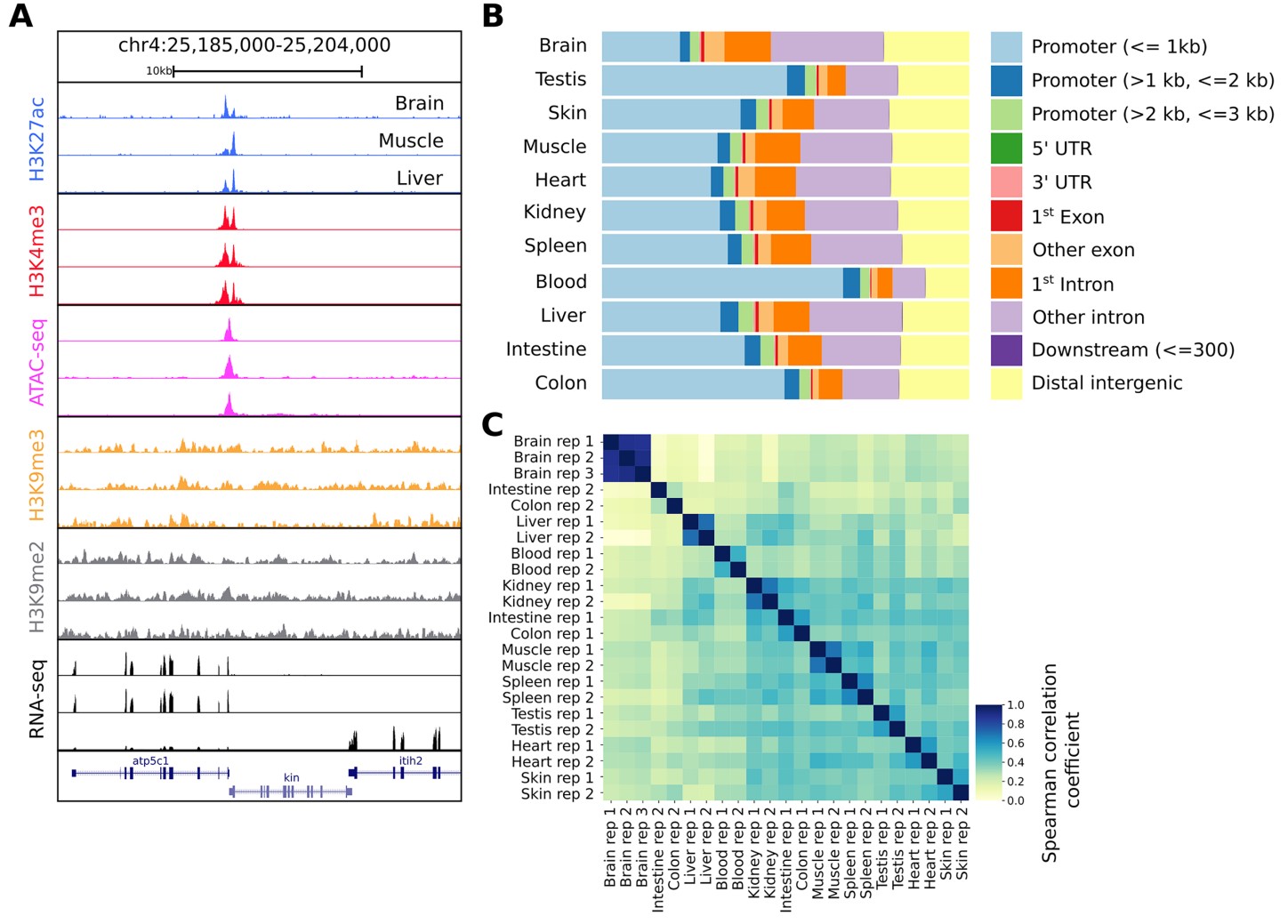

**Figure 2** Snapshot of the UCSC trackhub and quality figures of seq2science. (A) Fully configured UCSC trackhub generated by seq2science which highlights some of the supported assays. (B) The fraction of ATAC-seq peaks predicted in each tissue and their genomic distribution, visualized by ChIPseeker. (C) Pairwise Spearman correlation coefficients of all the samples.

that do not have a readily available motif annotation. Figure 3 shows the top motifs per tissue, based on the z-score. The complete table is part of the seq2science output report (see File S7). In general, this unsupervised motif analysis recapitulates many of the findings of *Yang et al. (2020)*, such as RFX and bHLH (Neurod, Atoh1) motifs enrichment in the brain and Hnf4a enrichment in the liver, colon, and intestine. Additionally, the GimmeMotifs analysis assigns Tp63 as a transcription factor enriched in the skin, as well as Tfap2, which are well-known regulators of epidermal development (*Soares & Zhou, 2017*; *Li et al., 2019*).

## The regulatory landscape of whole-body regeneration in the three--banded panther worm *Hofstenia miasma*

The article "Acoel genome reveals the regulatory landscape of whole-body regeneration" by *Gehrke et al. (2019)* analyzed the gene expression and chromatin accessibility of the Acoel worm *Hofstenia miamia* during regeneration. The article contains ATAC-seq and

**Figure 3 Result of the differential motif analysis by seq2science between ATAC-seq peaks of different tissues of zebrafish (*Yang et al., 2020*).** Per tissue, the top two most differentially enriched motifs have been selected, with automatically inferred orthologs. The transcription factor names were manually curated for clarity. The full table with the motif analysis results is included in the quality report (File S7).

RNA-seq time-series data of the response to amputation and during whole-body regeneration. These data, and the *Hofstenia miasma* genome are available from NCBI and consequently, seq2science can be applied to re-analyze this data with ease.
The configuration files to reproduce the seq2science analysis and the complete output and quality control report are provided as Supplemental Information (Files S8–S13).

The seq2science ATAC-seq workflow generated a consensus peak set based on the union of peaks from all time points, together with a count table with the read quantification per time point. See File S10 for the ATAC-seq QC report. We clustered the count table to visualize the temporal accessibility patterns using log-transformed and z-score normalized read counts (Fig. 4A). The figure nicely recapitulates the original work, with most ATAC-seq peaks showing strong signal at either 0 or 48 h post-amputation (hpa).

To demonstrate the RNA-seq workflow, we performed a differential expression analysis of 6 *vs.* 0 hpa using DESeq2 (see File S13 for the complete output). The results are visualized as a volcano plot in Fig. 4B. As reported, the *Hofstenia miamia* EGR ortholog is the most significant differentially expressed gene, followed by the RUNX homolog *runt*, which is involved in wound healing and regeneration. The role of *egr* is further confirmed by the differential motif accessibility analysis, performed on the ATAC-seq peaks in the seq2science ATAC-seq workflow (Fig. 4C). As stated earlier, GimmeMotifs automatically assigns the transcription factors in the database to the orthologous genes in our assembly.

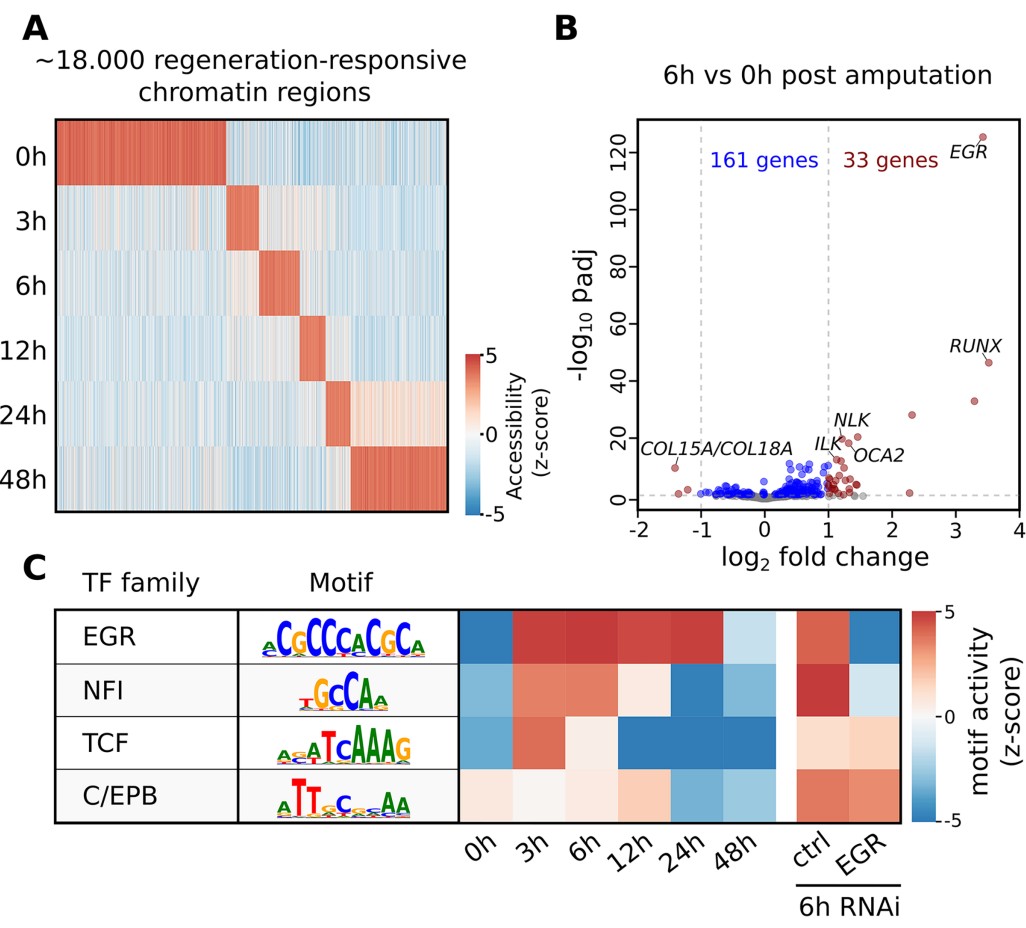

**Figure 4 Summary of selected results from the re-analysis of the RNA-seq and ATAC-seq regeneration time series from** *Gehrke et al. (2019)*. (A) Heatmap of *Hofstenia miamia* chromatin accessibility during tail regeneration post amputation. ATAC-seq read counts provided by seq2science were log-transformed, and columns were normalized using the z-score. (B) Differential gene expression during tail regeneration. The X axis shows the log2 fold change of 6 hpa *vs.* 0 hpa; the Y axis shows the −log10 transformed *p*-value. Significantly changed genes (padj ≤ 0.05; log2 fold change ≥ 1) are marked in red. Top results were labeled with human ortholog gene families. (C) Motif activity prediction during tail regeneration. The top four motifs, as identified by GimmeMotifs, are shown with the activity z-score predicted by gimme maelstrom. Both *Gehrke et al. (2019)* and our seq2science analysis identified the EGR and NFI motifs as the top differentially active motifs in the knockdown experiment.

As expected, the top enriched motif is EGR, which shows a high z-score during the post-amputation time-series, from 3 to 24 hpa. After the knockdown of EGR using RNAi, this is the most depleted motif with a strong negative z-score.

In conclusion, this use case reproduced the main findings from the the original paper, using the default tools incorporated in the RNA-seq and ATAC-seq workflows. Additionally, it demonstrates that seq2science is also easily applicable to non-model species.

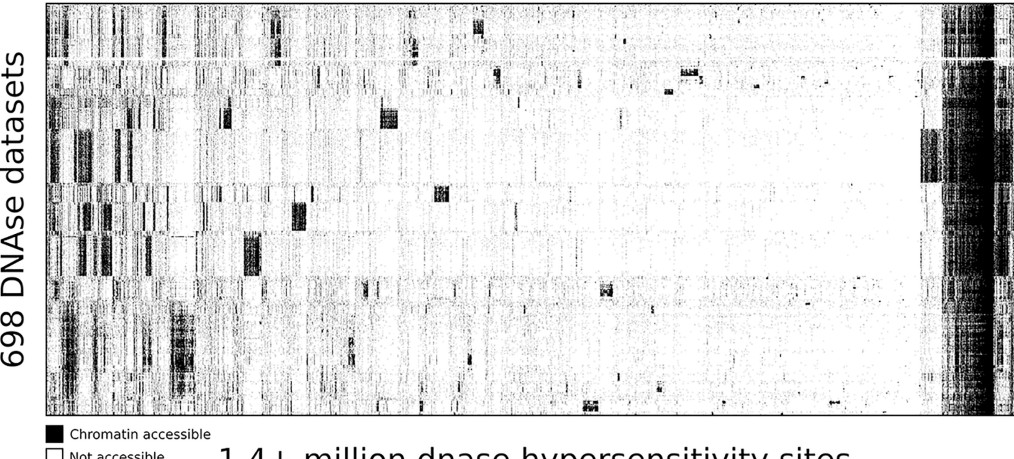

**Figure 5** **DNA accessibility at 1.4 million consensus DHSs assayed across 698 samples encapsulated in a visually compressed DHS-by-biosample matrix.** Recurring accessibility patterns indicate extensive sharing across cell contexts. Chromatin accessibility is defined as more than 1 count per million reads.

## Map of human DNase I hypersensitive sites

To highlight the ability of seq2science to scale with large data sets we performed a re-analysis of "Index and biological spectrum of human DNase I hypersensitive sites" (*Meuleman et al., 2020*). This article analyzed 733 human DNAse I samples across different tissues. Of the 733 samples they reported, we were able to find 698 in the SRA database. To accommodate the DNAse I assay, which is not supported by default, we used the high degree of customizability of seq2science and adapted the default ATAC-seq workflow by turning the tn5_shift flag off. The final output consists of over 1.7 TB of sorted BAM files, with 1,404,721 peaks in the consensus peak set between these samples. Figure 5 shows the clustered output of the final count table, where the distinction between accessible and not accessible is made based on whether there is more than 1 count per million reads or not.

## CONCLUSION

Seq2science facilitates reproducible preprocessing of high-throughput sequencing data of different assays through a unified setup. The tool is integrated with all major public sequence and assembly databases, and outputs extensive quality control and processed results to speed-up analysis. Seq2science requires minimal user input to get started, but offers a high degree of customizability. Each workflow and its configurable options are fully explained in the online documentation. The output from seq2science is reproducible and directly ready for analysis.

## ACKNOWLEDGEMENTS

We are grateful to the open source bioinformatics community for their support in the development of seq2science. Special thanks to the bioconda and conda-forge teams for their help with packaging. More specifically we want to thank Saket Choudhary, Johannes

Köster, Tao Liu, Devon Ryan, and Phil Ewels for their assistance with Pysradb, Snakemake, MACS2, Bioconda, and MultiQC respectively.

### Funding

This work was supported by the Netherlands Organization for Scientific Research (NWO Grant 016.Vidi.189.081). The funders had no role in study design, data collection and analysis, decision to publish, or preparation of the manuscript.

### Grant Disclosures

The following grant information was disclosed by the authors:
The Netherlands Organization for Scientific Research: 016.Vidi.189.081.

### Competing Interests

The authors declare that they have no competing interests.

### Author Contributions

- Maarten van der Sande conceived and designed the experiments, performed the experiments, analyzed the data, prepared figures and/or tables, authored or reviewed drafts of the article, software; Methodology, and approved the final draft.
- Siebren Frölich conceived and designed the experiments, performed the experiments, analyzed the data, prepared figures and/or tables, authored or reviewed drafts of the article, software; Methodology, and approved the final draft.
- Tilman Schäfers conceived and designed the experiments, performed the experiments, authored or reviewed drafts of the article, software; Methodology, and approved the final draft.
- Jos G. A. Smits conceived and designed the experiments, performed the experiments, authored or reviewed drafts of the article, software; Methodology, and approved the final draft.
- Rebecca Regina Snabel conceived and designed the experiments, performed the experiments, authored or reviewed drafts of the article, software; Methodology, and approved the final draft.
- Sybren Rinzema performed the experiments, authored or reviewed drafts of the article, software; Methodology, and approved the final draft.
- Simon J. van Heeringen conceived and designed the experiments, authored or reviewed drafts of the article, supervision, and approved the final draft.

### Data Availability

The software is available via the environment manager Anaconda and the seq2science package on bioconda (https://www.nature.com/articles/s41592-018-0046-7) and can be installed with "mamba install -c bioconda seq2science".
The source code is available at GitHub and Zenodo:

- https://github.com/vanheeringen-lab/seq2science
- Maarten van der Sande, Siebren Frölich, Jos Smits, Tilman Schäfers, Rebecca Snabel, & Simon van Heeringen. (2023). seq2science (v1.2.0). Zenodo. https://doi.org/10.5281/zenodo.8361102

The regularly updated documentation is available at: https://vanheeringen-lab.github.io/seq2science/.

The data is available at Zenodo: Maarten van der Sande, & Siebren Frölich. (2023). Seq2science manuscript supplementary data. Zenodo. https://doi.org/10.5281/zenodo.8345208

File S1

Installation instructions for seq2science.

https://zenodo.org/record/8345208/files/installation.txt

File S2

Configuration file for the alignment of ChIP-seq and RNA-seq samples of Yang et al. to the Danio rerio genome.

https://zenodo.org/record/8345208/files/zebrafish_alignment_config.yaml

File S3

Samples file for the alignment of ChIP-seq and RNA-seq samples of Yang et al. to the Danio rerio genome.

https://zenodo.org/record/8345208/files/zebrafish_alignment_samples.tsv

File S4

HTML quality control report including analysis figures for the alignment of ChIP-seq and RNA-seq samples of Yang et al. to the Danio rerio genome.

https://zenodo.org/record/8345208/files/zebrafish_alignment_QC.html

File S5

Configuration file for the alignment and analysis of the ATAC-seq samples of Yang et al.

https://zenodo.org/record/8345208/files/zebrafish_atac_config.yaml

File S6

Samples file for the alignment and analysis of the ATAC-seq samples of Yang et al.

https://zenodo.org/record/8345208/files/zebrafish_atac_samples.tsv

File S7

HTML quality control report including analysis figures for the ATAC-seq samples of Yang et al.

https://zenodo.org/record/8345208/files/zebrafish_atac_QC.html

File S8

Configuration file for the alignment and analysis of the ATAC-seq samples of Gehrke et al.

https://zenodo.org/record/8345208/files/acoel_atac_config.yaml

File S9

Samples file for the alignment and analysis of the ATAC-seq samples of Gehrke et al.

https://zenodo.org/record/8345208/files/acoel_atac_samples.tsv

File S10

HTML quality control report including analysis figures for the ATAC-seq samples of Gehrke et al.

https://zenodo.org/record/8345208/files/acoel_atac_QC.html

File S11

Configuration file for the alignment and analysis of the RNA-seq samples of Gehrke et al.

https://zenodo.org/record/8345208/files/acoel_rna_config.yaml

File S12

Samples file for the alignment and analysis of the RNA-seq samples of Gehrke et al.

https://zenodo.org/record/8345208/files/acoel_rna_samples.tsv

File S13

HTML quality control report including analysis figures and differential expression analysis for the RNA-seq samples of Gehrke et al.

https://zenodo.org/record/8345208/files/acoel_rna_QC.html

File S14

Directed acyclic graph of all the steps involved with the ATAC- and ChIP-seq workflow.

https://zenodo.org/records/8345208/files/DAG_genomic.pdf

File S15

Directed acyclic graph of all the steps involved with the RNA-seq workflow.

https://zenodo.org/records/8345208/files/DAG_rnaseq.pdf

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
