# Peer review of "Seq2science: an end-to-end workflow for functional genomics analysis"

_PeerJ, doi:10.7717/peerj.16380_

## Round 0.1 · original submission · Major Revisions

Dear Dr. van der Sande and colleagues:

Thanks for submitting your manuscript to PeerJ. I have now received three independent reviews of your work, and as you will see, the reviewers raised some concerns about the research. Despite this, these reviewers are optimistic about your work and the potential impact it will have on research studying -omics and informatics. Thus, I encourage you to revise your manuscript, accordingly, taking into account all of the concerns raised by both reviewers.

While the concerns of the reviewers are relatively minor, this is a major revision to ensure that the original reviewers have a chance to evaluate your responses to their concerns. There are many suggestions, which I am sure will greatly improve your manuscript once addressed.

Please ensure that all materials and methods are made available and thoroughly explained, with your workflow being repeatable. Please consider adding references deemed missing by the reviewers, and work on overall clarity and delivery of your paper such that a target audience is adequately defined.

I look forward to seeing your revision, and thanks again for submitting your work to PeerJ.

Good luck with your revision,

-joe

Reviewer 1 ·

Basic reporting

The manuscript by van der Sande et al describes seq2science, a powerful Snakemake-based workflow suite for the analysis of most common experiments in genomics. The manuscript focuses on bulk ATAC-seq, ChIP-seq and RNA-seq, although the Git repository contains additional functionality, primarily for analysing single cell experiments. The paper is written clearly and argues its case well. For pedagogical reasons, I would have preferred to have some workflow code included in the text itself, but good documentation could compensate for that.

Experimental design

I have been able to install seq2science (but see comment below) and re-run parts of the use cases. All I have tried works as it should. The software is designed and documented well. The workflow enables one to easily set up an analysis that starts from the retrieval of datasets from public databases, to their standardised processing and genome mapping, to calling signals in data, differential signal analysis, and browser track and track hub export. It all looks very powerful, and it is described in enough detail to repeat.

Validity of the findings

There is no question that the findings of the use cases are valid. The question is what its intended audience for the tool is. Even though it allows a high increase in productivity and reproducibility, it is primarily a tool for expert users. Users with limited exposure to workflows, let alone programming, would probably face a steep learning curve for anything other than pipelines that closely follow the provided use cases. This is not a complaint - experts, too, deserve tools to make their life easier (or at least more productive).

Additional comments

Minor comments:


- I know that is not the authors’ fault, but the primary reason I am late with this review is that seq2science is installed as part of bioconda. Sadly, full 2.5 years after Apple has started to ship computers with ARM processors, bioconda still does not support this platform, which makes it impossible to install anything from it unless I downgrade the entire Anaconda to the Intel version (which I do not want to do) or find another computer to do it (which took me several days to organise). I would urge the authors to exert pressure on bioconda admins/maintainers to start supporting Apples new (and by now not so new) CPUs - a lot of genomics analysis and education is done on them these days.

- Line 110: Explain briefly what ascp is - not everybody knows that, even among the targeted audience.

- Line 182: “to save compute.” - something is missing here.

- Line 220: “NBCI” - should be “NCBI”.

- Line 249: “1.404.721” - in all versions of English, comma, not period, is used as thousands separator; i.e. it should be “1,404,721”. Alternatively, use hard spaces.

·

Basic reporting

In this manuscript, van der Sande et al. describe ‘seq2science’, a collection of workflows and pipelines that aims to facilitate and standardise the analysis of a range of (epi-)genomic sequencing data, including single-cell experiments. Although a number of similar pipelines already exist, the authors claim that seq2science excels through ease of handling public data (from different sources) and data from non-model organisms. Seq2science only requires two input files, a pointer to fastq files and a configuration file of which a template can be generated through the ‘init’ functionality. As a proof of concept, the authors re-analyse a multi-omics study in zebrafish, a time-series experiment in the three-banded panther worm, and chromatin accessibility in a large number of human samples.

In general the manuscript is well written and easy to understand. Additionally, the accompanying readthedocs page is very elaborate, making the installation procedure and running the workflows straightforward. As stated by the authors, gathering data from different public resources is effortless and starting the workflows is a matter of minutes.

Experimental design

The research fits within PeerJ’s scope and has a clearly defined research question. The seq2science workflow is well documented within the manuscript (and on the readthedocs page). The use cases are relevant to the research question (integration of multi-omics datasets, usage of a non-model organism, and processing a large number of samples) and are well documented so the analyses are reproducible.

It would be nice if the authors could elaborate on three items:

a) As stated in the manuscript: the differential motif analysis is based on orthology (which, as stated, is useful for non-model organisms). It is not 100% clear what this means. Can motif analyses be done on organisms that only have gene annotations available ? And vice-versa, can motif analyses be done using motif annotations ? Perhaps this part could be elaborated on.

b) From the documentation, batch effects can be accounted for in the implemented differential tests (DESeq2) by adding them as a factor in the design. This deserves some explanation in the manuscript as well, as batch effect correction is important in the analyses of (multiple) public datasets. It would also be useful to know if batch correction is only implemented for differential tests, or for example in sample normalisation as well.

c) Differential testing with contrasts is also explained in the readthedocs. Perhaps this could be elaborated as well (either in the docs or in the manuscript):

- Are all the samples included in the DESeq2 object or only those specified in the relevant contrast ? This could have an effect on dispersion calculations and thus the end result.

- Relevant to the previous point: should one split up multiple runs when samples are confounded with a batch effect ?

Validity of the findings

The findings and conclusions are valid, supported and appropriately reported.

Additional comments

There are only few additional (minor) comments:

- ‘Seq2science init’ doesn’t print out the ‘valid workflows’ which it requests. One has to go through the documentation to find these. It would be nice if these can be printed when invoking init with either no arguments, or with -h / –help.

- ‘Seq2science run’ has a –profile argument. Presumably this is where the profile to submit to a compute grid would be specified (qsub / slurm / …). The URL in the CLI (https://github.com/s2s-profiles) points to a private (?) repository. Could this be updated (potentially to https://github.com/vanheeringen-lab/seq2science-profiles)? If these are just snakemake profiles (rather than seq2science profiles), one could just link to snakemake’s profile repository (https://github.com/Snakemake-Profiles).

- The wide range of public data that is accessible is already impressive. The authors may want to consider also the following: ENCODE: (https://www.encodeproject.org), IHEC (https://epigenomesportal.ca/ihec) or EpiRR (https://www.ebi.ac.uk/epirr)
Those present large efforts to structure epigenome data sets and make them searchable. It is likely that these well-organised portals present a natural starting point for many large comparative analyses that seq2science would be able to support.

The review was submitted by:
Ward Deboutte
Thomas Manke

·

Basic reporting

# Review seq2science

Standardized analysis workflows that are easy to use, are a pre-requisite for (more) reproducible scientific results based on large datasets. This is especially important for genomic data in bio-medical research. The seq2science project and this manuscript address important issues in the creation, maintenance and use of such workflows. The authors provide a modular framework with re-usable building blocks that cover everything from automated download of public data for re-analysis to specific data analyses for a number of functional genomics data types. In addition, they created a command-line tool for quick deployment of their workflows, providing good default settings and extensive documentation. Altogether, this is an impressive project that definitely warrants publication, with only minor edit suggestions on my part.

## 1. Basic Reporting

The manuscript is well structured, with a good introduction to the necessary background. The presented use cases are well-documented within the manuscript and the supplementary files are available via Zenodo. All figures are relevant to the manuscript and well-labelled. The presented workflows can directly be run by downloading relevant data from public databases, and sample sheets for the use cases in the manuscript are provided.

Some very minor points, roughly in decending order of importance:

* I would suggest citing the newer Snakemake paper (instead of the old 2012 paper), which has a lot more information about the current feature set of Snakemake. It can be found here: https://doi.org/10.12688%2Ff1000research.29032.2
* Figure 1 is never cited in the text. But it does give a good indication of what the workflows can do, so maybe it warrants some explanation in the text.
* Related to the previous point, it might make sense to give an overview of what the individual use case workflows in the manuscript actually do. To this end, I would suggest adding an exemplary `--rulegraph` for at least one of the example workflows in the manuscript. And maybe also add `--rulegraph`s for all the other use case workflows in the supplement? A command to quickly produce them with snakemake should be: `snakemake --forceall --rulegraph | dot -Tpdf > dag.pdf`
* line 226: The reference should be to Figure 4A, not Figure 4B.
* Figure 2 B: It is unclear to me what the difference between the "Promoter" categories is? Or to put it differently, what does (1-2 1kb) mean? Is this supposed to be (1-2 kb), or even more precisely (>1, <=2 kb)?
* line 220, typo: NBCI -> NCBI

Experimental design

## 2. Experimental design

The project behind the manuscript is well motivated in the text. The use cases are diverse and very well explained, including the replication of results from the respective data sets' original publications. The seq2science project is very well documented online.

I have a couple of questions for the authors about their choice not to use certain snakemake features. This is mostly out of curiosity, but I think giving their motivation for these choices in the manuscript might make it even more useful for the snakemake user and developer community. Also, if the answer is something like "the feature was not available at the time of development" or "we did not find this / become aware of this feature" are absolutely valid, and would provide feedback to for example improve documentation and/or visibility of certain features. So, here go the questions:

* Why use only multiqc reports, which are mainly aimed at quality control output, for the full workflow output reporting? Snakemake provides a reporting functionality by itself: https://snakemake.readthedocs.io/en/stable/snakefiles/reporting.html
Furthermore, the authors might be interested in the datavzrd project: https://github.com/datavzrd/datavzrd
It provides automated rendering of interactive tables from TSV data, and the resulting HTML tables can easily be integrated into snakemake's reports.
* The authors put considerable efforts into solving common problems during workflow development. One of them is the impressive collection of samples2metadata_* functions, that help automating the download and handling of sequencing data from public repositories. It would be great to have such functionality available in snakemake wrappers, as this would then allow broader re-use: not only within seq2science, but also in any other snakemake workflow. Also, these wrappers are generally meant for exactly this purpose, abstracting complicated tasks or interfaces into copy-pastable snakemake rules. For details, see here: https://snakemake-wrappers.readthedocs.io/en/stable/
* The authors created their own deployment mechanism for seq2science workflows with `seq2science init {workflow}`. This can also be done with the `snakedeploy` command line utility. Any particular reasons for not using it? For documentation, see:
https://snakedeploy.readthedocs.io/en/latest/
* The authors created nicely modular workflows that can be chained together by running them consecutively (for example separate workflows to first download data, then do alignment, and finally one of the analysis workflows). Nowadays, snakemake natively supports such a modularization via its module import for modularization, which would also allow importing all of the sub-module workflows, configuring them in one configuration file, and then running them all at once. But probably this functionality didn't exist when the seq2science project was started? For documentation, see:
https://snakemake.readthedocs.io/en/latest/snakefiles/modularization.html#modules
* In addition, relying on snakemake wrappers for common rules and snakemake modules for modularization, would make it possible to easily include seq2science in the snakemake workflow catalogue and have all the sub-workflows deployable via the `snakedeploy` command. For an actively used and maintained example of such a standardized workflow, see:
https://github.com/snakemake-workflows/dna-seq-varlociraptor

Please note that I am open to being contacted by the authors, should they have questions about any of the mentioned projects / mechanisms.

Validity of the findings

## 3. Validity of the findings

All points relevant to this part have been covered in my comments on basic reporting and experimental design.

Additional comments

## 4. General comments

Some further suggestions and comments for the seq2science project:
* I would recommend updating the installation recommendations to miniforge (https://github.com/conda-forge/miniforge#install) as an installation of mamba, which is a drop-in-replacement for conda that can greatly speed up the dependency resolution for and creation of conda environments.
* The only place I could find a list of the available `{workflow}` names that can be used with `seq2science init` was in that command's error message when trying it out with some non-existent workflow name. I think the workflow name should be stated directly in the workflow docs online, and the `seq2science init` help message should list the available options. In addition, the getting started section of the docs could contain a table with all the available workflows and their names, although manually maintaining that might run the risk of having this out-of-sync with the actually available workflows.
* zebrafish_alignment_QC.html#samplesconfig: The Samples & Config table has the `sample` and `assembly` columns in the header, but the content is missing. Instead, it splits the comma-separated color column values across the last three columns. So something is off with the TSV parsing and display, here.

### Editing suggestion

* abstract sentence was:
Furthermore, it supports any genome and automates genome assembly retrieval from Ensembl, NCBI and UCSC.
* alternate suggestion:
Furthermore, it automates the retrieval of any genome assembly available from Ensembl, NCBI or UCSC.

---

## Round 0.2 · accepted · Accept

Dear Dr. van der Sande and colleagues:

Thanks for revising your manuscript based on the concerns raised by the reviewers. I now believe that your manuscript is suitable for publication. Congratulations! I look forward to seeing this work in print, and I anticipate it being an important resource for groups studying -omics and informatics. Thanks again for choosing PeerJ to publish such important work.

Best,

-joe

·

Basic reporting

NA

Experimental design

NA

Validity of the findings

NA

Additional comments

The authors have addressed all concerns. Their workflow is very useful to access and analyse large public data sets in a standardised manner. The newly added support for ENCODE data is very valuable.

The review was submitted by:
Ward Deboutte
Thomas Manke

·

Basic reporting

I thank the authors for thoroughly considering and addressing all reviewer comments, and recommend this manuscript for publication in this revised form.

Experimental design
* * *
Validity of the findings